# How to Promote Healthier and More Sustainable Food Choices: The Case of Portugal

Daniel Francisco Pais [1,*] , António Cardoso Marques [1] and José Alberto Fuinhas [2]

1 Research Unit in Business Science and Economics (NECE-UBI), Management and Economics Department, University of Beira Interior, Rua Marquês d'Ávila e Bolama, 6201-001 Covilhã, Portugal
2 Centre for Business and Economics Research (CeBER), Faculty of Economics, University of Coimbra, 3004-512 Coimbra, Portugal
* Correspondence: daniel.pais@ubi.pt

**Abstract:** The demand for food has been increasing throughout the years, with notable preferences for animal-based foods. Considering the impact of the excessive animal-based consumption on the environment and public health, international organisations and the scientific literature have advised for a large-scale transition towards healthier and more sustainable food consumptions, i.e., a systematic decrease in animal-based consumption followed by an increase in plant-based consumption. However, to effectively promote healthier and more sustainable food choices such as plant-based ones, it is crucial to understand what motivates consumers' food choices. Based on primary data (N = 1040), representative of the Portuguese population, it was possible to assess the potential motivators behind food choices, allowing to provide guidelines for policy decision. The impact of different socioeconomic characteristics, food consumption orientations, and food-related behaviours on food choices was estimated. In general, most of the drivers of plant-based meals were also motivators for reducing animal-based meals. The main findings demonstrate that the more environmentally conscious and informed the consumer, the more likely they are to choose more plant-based and less animal-based meals on a weekly basis; not only informed consumers, but consumers who actively look for information before buying choose more plant-based meals. Thus, not only information, but, more importantly, education regarding food characteristics and its impact on society should be the focus of policymakers. Understanding the drivers and barriers of food choices is vital for informing future food policy to promote healthier and more sustainable choices rich in plant-based foods, both for Portugal as well as for other European countries, particularly the southern ones with similar culture and where the Mediterranean diet is highly promoted.

**Keywords:** food economics; food choices; sustainable development; food education; primary data; logistic regressions

## 1. Motivation

Driven by population growth and the increase in consumers' purchasing power, the growing demand for food and how it is met is threatening the sustainability of present and future generations. Due to the growing demand, major changes in the food systems, from the last half a century, have aggravated both global health and sustainability challenges [1]. Following the established Bennett's law and Popkin's nutrition transition theory, the generalized and continuous increase in income, throughout the last half a century, has contributed to the rise of the demand for food, both total and per capita, and particularly for animal-based foods [2–4]. According to Food and Agricultural Organisation (FAO) data, animal-based consumption growth rates vary across countries, where high-income countries begin to show stagnation in recent years, although middle-income countries are reporting strong increases [5], as shown in Figure 1.

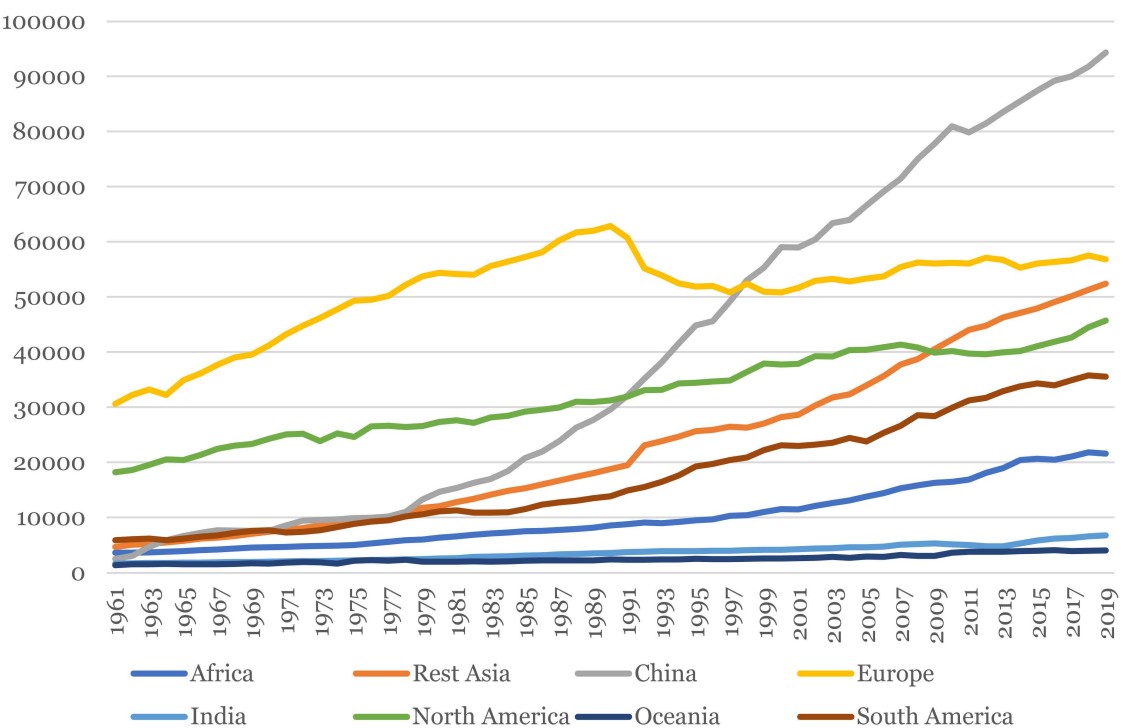

**Figure 1.** Animal-based consumption among different world regions (tonnes).

Following Figure 1, although it has plateaued, animal-based consumption is reaching its peak in high-income regions (Europe and North America) and shows a steep increase in middle-income regions (Asia and South America), particularly from China, which has tripled its total consumption in the last 30 years This rapid increase in animal-based consumption is associated with substantial effects on public health and can have major negative net effects on the environment and overall sustainability [6].

To achieve the targets proposed to mitigate climate change, particularly keeping global warming to below 2 °C, the Intergovernmental Panel on Climate Change (IPCC) suggests a shift in dietary habits [7]. According to Chapter 5 (Food Security) from the special report, where the mitigation potential of changing diets is analysed, diets with higher shares of animal-based foods are also associated with lower mitigation potentials, i.e., it suggests a direct negative relationship between the share of animal-based foods in the diet and its mitigation potential [8]. Additionally, Ruett et al. [9] demonstrate that plant-based diets and diets with low animal-based food intake could contribute to keeping global warming levels below the 2 degrees threshold. The authors conclude that increasing plant-based intake while reducing animal-based foods is a powerful strategy to achieve the targets proposed.

Furthermore, Clune et al. [10], Poore and Nemecek [11], and Reinhardt et al. [12], reviewing life cycle assessments, also conclude that animal-based foods tend to have higher ecological footprints (natural resources' requirements and emission of greenhouse gases) compared to plant-based foods. The current intensive livestock industry is considered one of the main drivers of land degradation, deforestation, clean water scarcity, loss of biodiversity, and climate change [13–15]. Chai et al. [16] present a literature review on the environmental impacts of different diets, suggesting that the plant-based diet is the optimal diet for the environment due to its lowest greenhouse gas emissions. In Portugal, food consumption is the main driver (30%) of its ecological footprint [17]. As a consequence, for 2016, if everyone lived like an average Portuguese citizen, 2.3 planets earth would be required, in terms of resources, to satisfy a year of consumption [18]. In terms of national resources, for 2018, Portugal requires 3.52 times its current annual resources capacity to satisfy a year of consumption (see also Global Footprint Network [19]).

The IPCC special report further concludes that food choices can effectively help achieve mitigation targets and that low-carbon diets, mainly plant-based, on average, tend to be

healthier and with smaller land footprints. Not only can plant-based foods be more sustainable, but they also tend to generally be healthier, compared to animal-based foods [20]. Nelson et al. [21], Chen et al. [22], and Springmann [23] also conclude that diets mainly composed of plant-based foods, with low shares of animal-based foods, tend to be both healthier and more sustainable. From a public health perspective, although animal-based foods are nutritious, as they provide some essential macro- and micronutrients, today, it is possible to satisfy such nutritional requirements without animal-based foods if a planned and diversified diet is followed, the Academy of Nutrition and Dietetics concludes [24].

The current global diet is generally associated with increases in non-communicable diseases [25,26], including coronary heart disease, the world's leading cause of death [27]. This relationship is particularly seen in consumers following diets with high levels of red and processed meat [28–30], with the consumption of processed meat having a higher impact on human health than unprocessed meat [31,32]. Furthermore, according to the World Health Organisation (WHO), processed meat was classified as "carcinogenic to humans" and red meat "probably carcinogenic to humans", which warns about the potential negative effects of the current increase in demand for these types of meat [28], which the literature also corroborates [33,34]. In particular, Bradbury et al. [35] suggest that consuming red and processed meat at an average of 76 g per day was associated with an increased risk of colorectal cancer.

Considering the above, the Lancet Commission presented an extensive literature review supporting the need for a substantial dietary shift, which includes a drastic reduction in global consumption of animal-based foods, particularly red meat, and a greater than 100% increase in plant-based foods, particularly fruits and vegetables [36]. The authors suggest that such dietary shift could prevent approximately 11 million deaths per year globally and could sustainably produce enough food for the growing population without further damaging the environment. Specifically, they recommend consuming red meat no more than 14 g/day, and a total meat consumption of no more than 75 g/day, updating older recommendations [37]. The scientific literature also strengthens these claims, supporting a demand shift away from current resource-intensive foods toward a greater reliance on more sustainable and healthier ones, particularly plant-based [38–40]. Consequently, attention is increasing on understanding food choices to allow for the development of effective policies and strategies in reducing animal-based consumption and increasing plant-based foods [41–43]. Galli et al. [44] suggest that changing consumers' food choices in Portugal, particularly through major reductions in animal-based consumption, could lead to a decrease in Portugal's ecological deficit up to 19%, that is, less resources would be required to satisfy the same nutritional needs and thus the ecological deficit would decrease.

Therefore, it is imperative to better understand the factors that influence consumers' food choices to develop effective interventions and policies. The challenge of how to promote healthy and sustainable diets in ways that appeal to increasingly greater numbers of consumers, that is, in an effective and efficient way, is only possible if knowledge on consumers' motivations behind food choices is available and well understood; not only consumers' preferences, but the relationship these have with current food choices. With this knowledge, it will be possible to materialize potential policies, from targeted strategies, novel products, educational campaigns, and materials, among others, to promote healthy and sustainable diets.

The literature shows some development in understanding consumer preferences and their relationship with food choices to assist in addressing this challenge. Using aggregated data, Milford et al. [45] discuss the drivers of meat consumption, whereas Pais et al. [46] analyse the drivers of a dietary transition towards plant-based diets. The authors identify income, urbanization, prices, education level, and globalization as potential drivers. Moreover, many are the frameworks in which consumer preferences are analysed via online questionnaires. Dominici et al. [47], through logistic regressions using online questionnaire data in Italy, investigate the factors that influence consumers' likelihood to buy food online. The authors show that some motivators, such as being obese, working

time, and having health problems, positively affect the probability of buying food online, whereas car possession and the distance from home-to-store do not influence online shopping. Ferreira et al. [48] assess the impact of region of origin on food choices related to wine consumption in Portugal, whereas Kilders et al. [49] assess the impact of consumer ethnocentricity on food choices related to imported foods in Nigeria. Consumer preferences associated with animal welfare are the focus of Cao et al. [50], concerning food choices related to egg consumption in Canada. More recently, there has been interest in understanding the impact of the COVID-19 pandemic on food choices, specifically, the effect on food security using online questionnaires [51–53].

Other authors have explored potential drivers around changing food choices, particularly, reducing animal-based consumption. Edenbrandt and Lagerkvist [54] assess the effectiveness of food labelling in changing food choices in Sweden, particularly, in reducing and substituting animal-protein sources. The authors conclude that information concerning the environmental impact of foods (traffic-light carbon labels) positively affects food choices by increasing the willingness to purchase lower-emissions foods such as poultry and meat substitutes. This effect is largest among consumers who already follow sustainable food choices. Following a representative sample of the Danish population, Hielkema and Lund [55] explore which food-related factors may act as drivers or barriers toward the reduction of meat consumption. Corroborating with the latter, the authors found that knowledge regarding the environmental impact of foods, particularly meat, is a driver of meat reduction (see also Hunter and Röös [56]). Potential barriers include habitual behaviour, food neophobia, and identity incongruence. The authors conclude that it is vital that strategies focus on meat reduction rather than total exclusion, as completely removing meat from diets was regarded as unpopular. For a comprehensive review on potential motivators behind food choices, see both Graça et al. [57] and Enriquez and Archila-Godinez [58].

Considering the importance of understanding which motivators can be exploited to promote healthy and sustainable diets effectively, the present study contributes to help materialize the dietary transition needed through the following goals: (i) identify consumers' current food choices; (ii) build a consumer profile which includes socioeconomic characteristics, food orientations, preferences, and behaviours; and (iii) assess the relationship between the potential factors that constitute the consumer profile and current food choices, analysing the motivators (drivers as well as barriers) behind the different food choices identified. By understanding what motivates food choices, it will be possible to derive effective food policy implications to promote healthier and more sustainable diets, particularly, more plant-based rich diets and less animal-based ones as the literature suggests. To achieve this, the analysis uses a representative sample of Portuguese consumers (N = 1040) collected via an online questionnaire.

The main findings suggest that plant-based meals are primarily motivated by orientations regarding animal welfare, the environment, and naturalness. Conscious, more informed, and aware consumers tend to eat more plant-based meals and are associated with less animal-based meals. In general, most of the drivers of plant-based meals are also associated with a reduction of animal-based meals. Food policy should focus not only on promoting clear and accessible information, but more importantly, on educating consumers to be more conscious (of their actions' impact) and informed when consuming. Accessible information will only be efficient if consumers seek and know how to use it. Thus, education is vital to reroute the current rooted dietary habits toward healthier and more sustainable ones. Other policy implications are presented and discussed within the literature.

## 2. Data and Methods

### 2.1. Respondents and Sample Procedure

Data for the present study were collected through a cross-sectional online questionnaire. The questionnaire was constructed based on recent literature on food choices, particularly, the Food Demand Survey (FOODs) [59], currently Meat Demand Monitor (for

additional information, including reports, raw data, and survey instrument files, consult Kansas State University's website: https://www.agmanager.info/livestock-meat/meat-dem and/monthly-meat-demand-monitor-survey-data (accessed on 26 December 2022)), which administers a monthly online questionnaire concerning food choices, preferences, and views for the U.S. population [60]; the Food Choice Questionnaire [61]; and the questionnaire from Graça et al. [62]. Through a period of 3 months starting on 12 February 2021, a total of 2332 completed responses were collected. Of these, 204 were discarded due to inconsistencies in the answers throughout the questionnaire. This led to a total sample size of 2128. Afterward, to guarantee a representative sample of consumers in Portugal, quota sampling was conducted, considering gender, age groups, geographical region, and educational attainment. Thus, a nationally representative subsample of active Portuguese consumers was constructed using an algorithm that randomly chose responses with the constrain of optimizing the representativeness considering the demographic characteristics commonly used in the literature. The subsample, which will be the focus of the present study, contains a total of 1040 responses broadly representative of the Portuguese population in terms of gender, age, and region; however, it leans toward a slightly more educated demographic, which is common in online-based questionnaires, as shown in the literature [50,63,64]. Demographic details of the subsample are provided in Table 1.

**Table 1.** Demographic characteristics of responses.

|  | **Sample (2128)** | **Subsample (1040)** | **Portugal *** | **Δ **** |
|---|---|---|---|---|
| *Gender* |  |  |  |  |
| Female | 75.05% (1597) | 52.88% (550) | 51.75% | 1.39 |
| Male | 24.72% (526) | 46.63% (485) | 48.25% | −1.61 |
| Non-binary | 0.23% (5) | 0.48% (5) | - | - |
| *Age groups* |  |  |  |  |
| 15–19 | 8.22% (175) | 8.27% (86) | 8.24% | 0.03 |
| 20–24 | 20.44% (435) | 11.06% (115) | 8.32% | 2.74 |
| 25–29 | 13.06% (278) | 8.37% (87) | 8.27% | 0.10 |
| 30–34 | 9.45% (201) | 8.56% (89) | 8.56% | 0.00 |
| 35–39 | 9.30% (198) | 10.19% (106) | 10.16% | 0.03 |
| 40–44 | 10.57% (225) | 11.83% (123) | 11.85% | −0.02 |
| 45–49 | 10.15% (216) | 11.92% (124) | 11.93% | −0.01 |
| 50–54 | 7.89% (168) | 11.25% (117) | 11.26% | −0.01 |
| 55–59 | 7.28% (155) | 11.15% (116) | 11.18% | −0.03 |
| 60–64 | 3.62% (77) | 7.4% (77) | 10.23% | −2.83 |
| *Regions* |  |  |  |  |
| Norte | 41.02% (873) | 35.87% (373) | 35.88% | −0.01 |
| Centro | 21.33% (454) | 21.25% (221) | 21.25% | 0.00 |
| Lisboa | 19.64% (418) | 26.83% (69) | 26.84% | 0.02 |
| Alentejo | 7.14% (152) | 6.63% (279) | 6.61% | 0.01 |
| Algarve | 4.61% (98) | 4.13% (43) | 4.18% | −0.05 |
| R.A. Açores | 2.63% (56) | 2.6% (27) | 2.56% | 0.04 |
| R.A. Madeira | 3.62% (77) | 2.69% (28) | 2.69% | 0.00 |
| *Education* |  |  |  |  |
| Higher education | 70.21% (1494) | 59.23% (616) | 25.38% | 33.85 |

Notes: * Census-estimated data concerning the year of 2019 extracted from Statistics National Institution (www.ine.pt). ** Difference between representative subsample and population.

The subsample size (N = 1040) is due to the population size (approximately 6.619 thousand Portuguese consumers between the age of 15 and 64 in 2019), and a sampling error of 4% with a 99% confidence interval, which was calculated with the following formula:

$$\text{Sample size} = \frac{\frac{z^2 * p(1-p)}{e^2}}{1 + \left(\frac{e^2 * p(1-p)}{e^2 * N}\right)},$$

where $z$ denotes the $z$-score, which is the number of standard deviations a given proportion is away from the mean related with the desired confidence level, $e$ denotes the margin of error in decimal form, and $N$ denotes the population size to be assessed.

### 2.2. Variables and Model Specification

The present study uses primary data collected from an online questionnaire on food choices (a copy of the questionnaire is available in the Supplementary Material). The dependent variables correspond to the frequency of meals containing red meat (*MRED*), white meat (*MWHT*), fish (*FISH*), ovo-lacto-vegetarian meals (*OLVG*), and vegan meals (*VEGA*). Thus, a total of five dependent variables, summarized in Table 2, were used. These variables were coded as ordinal with four categories: (a) zero meals, (b) one to three meals, (c) four to six meals, and (d) seven or more meals consumed per week.

**Table 2.** Summary of the dependent variables.

| Variable | Definition | Mean | Std. Dev. | Min. | Max. |
|----------|-----------|------|-----------|------|------|
| *MRED* | Red meat meals per week | 2.202 | 0.715 | 1 (zero meals) | 4 (seven or more meals) |
| *MWHT* | White meat meals per week | 2.642 | 0.830 | 1 (zero meals) | 4 (seven or more meals) |
| *FISH* | Fish meals per week | 2.440 | 0.710 | 1 (zero meals) | 4 (seven or more meals) |
| *OLVG* | O-L-Vegetarian meals per week | 2.759 | 0.814 | 1 (zero meals) | 4 (seven or more meals) |
| *VEGA* | Vegan meals per week | 2.277 | 1.032 | 1 (zero meals) | 4 (seven or more meals) |

The potential motivators assessed from the questionnaire are presented in four groups: (1) the socioeconomic characteristics; (2) general food consumption orientations; (3) specific food consumption orientations and concerns; and (4) food consumption preferences and behaviours. A total of 51 variables were assessed. These are summarized in Table 3, where their definitions and brief descriptive statistics are shown (a detailed description of all variables is available in the Supplementary Material).

Following Table 3, group 1 of the socioeconomic characteristics consists of 16 variables, from continuous (*AGE*, *BMI*, *HRS*) to ordinal (*INC*) and binary (*FEM*, *EDU*, etc.) variables. Group 2, concerning the general food consumption orientations, consists of 15 variables where the respondent was asked "Select which of the following factors are more important to you when buying food". A similar question was conducted for Group 3 (9 variables): "As a consumer, my concerns regarding food are ... ". Both Groups 2 a 3 use a 5-point Likert-type scale for answers. Finally, food consumption preferences and behaviours (Group 4) include 11 variables, from binary (*INFO*, *COOK*, etc.) to ordinal (*FEXP*) and continuous (*FAFH*, *FRTE*) variables. Additionally, a summated scale variable based on 15 items was constructed, measuring the rate of consumers' awareness about some food-related issues (*AWAR*). The items used are described in Table S1, in the Supplementary Material. The Cronbach's alpha for the 15 items is 0.886, suggesting that the items have relatively high internal consistency ($0.7 < \alpha < 0.9$) [65].

To isolate the relationship between the potential motivators and the likelihood of consumers choosing different meals, four models using ordinal logit regressions (non-linear) were performed for each type of food choice. Therefore, the general specification of the models for each are as follows:

$$MEALS = f(AGE; FEM; EDU; SEA; BMI; SNG; FAM; K12; STD; HRS; INC; LFT; RGT; CNS; LIB; CTL) \quad (1)$$

$$MEALS = f(APE; \; AWL; \; PLE; HLT; NVL; ORG; NTR; ENV; PRC; FRT; CNV; NAT; STQ; VRT; INF) \quad (2)$$

$$MEALS = f(CHLD; LWGT; GWHT; TIME; AVOI; LBLS; LOCA; PRNC; CONX) \quad (3)$$

$$MEALS = f(INFO; BIOL; COOK; FAFH; FRTE; LFTO; SHOP; FEXT; ROFP; CHNG) \quad (4)$$

Overall, a total of 20 models were estimated. For all, a Variance Inflation Factor (VIF) below 10 was observed, not raising concerns of eventual multicollinearity, and the residuals showed no marked deviations from normality. All likelihood ratios were statistically significant at the 1% level, securing the model's consistency.

**Table 3.** Summary of potential motivators.

|  | Variable | Definition | Mean | Std. Dev. | Min. | Max. |
|---|---|---|---|---|---|---|
| Group 1 | AGE | Age | 39.939 | 13.786 | 15 | 64 |
|  | FEM | 1 if female | 0.531 | 0.499 | 0 | 1 |
|  | EDU | 1 if higher education | 0.592 | 0.492 | 0 | 1 |
|  | SEA | 1 if seaside | 0.768 | 0.422 | 0 | 1 |
|  | BMI | Body mass index | 24.961 | 4.532 | 13.84 | 44.19 |
|  | SNG | 1 if single | 0.456 | 0.498 | 0 | 1 |
|  | FAM | 1 if family | 0.861 | 0.347 | 0 | 1 |
|  | K12 | 1 if kids under twelve | 0.252 | 0.434 | 0 | 1 |
|  | STD | 1 if student | 0.242 | 0.429 | 0 | 1 |
|  | HRS | Working hours | 31.904 | 16.546 | 0 | 90 |
|  | INC | Disposable income | 2.630 | 1.243 | 1 | 7 |
|  | LFT | 1 if left | 0.311 | 0.463 | 0 | 1 |
|  | RGT | 1 if right | 0.170 | 0.376 | 0 | 1 |
|  | CNS | 1 if conservative | 0.063 | 0.244 | 0 | 1 |
|  | LIB | 1 if liberal | 0.302 | 0.459 | 0 | 1 |
|  | CTL | 1 if catholic | 0.598 | 0.491 | 0 | 1 |
| Group 2 | APE | Appearance | 3.566 | 1.075 | 1 | 5 |
|  | AWL | Animal Wellbeing | 3.453 | 1.089 | 1 | 5 |
|  | PLE | Pleasure | 4.350 | 0.709 | 1 | 5 |
|  | HLT | Health | 4.367 | 0.780 | 1 | 5 |
|  | NVL | Novelty | 2.559 | 1.065 | 1 | 5 |
|  | ORG | Origin | 3.362 | 1.175 | 1 | 5 |
|  | NTR | Nutrition | 3.975 | 0.938 | 1 | 5 |
|  | ENV | Environment | 3.701 | 1.021 | 1 | 5 |
|  | PRC | Price | 4.044 | 0.831 | 1 | 5 |
|  | FRT | Fairtrade | 3.907 | 0.889 | 1 | 5 |
|  | CNV | Convenience | 3.802 | 0.870 | 1 | 5 |
|  | NAT | Naturalness | 3.875 | 0.958 | 1 | 5 |
|  | STQ | Status quo | 1.911 | 1.060 | 1 | 5 |
|  | VRT | Variety | 4.041 | 0.849 | 1 | 5 |
|  | INF | Information | 3.822 | 0.940 | 1 | 5 |
| Group 3 | CHLD | Food for children | 3.164 | 1.415 | 1 | 5 |
|  | LWGT | Losing weight | 3.225 | 1.289 | 1 | 5 |
|  | GWGT | Gaining weight | 2.178 | 1.325 | 1 | 5 |
|  | TIME | Finding time | 3.638 | 1.068 | 1 | 5 |
|  | AVOI | Avoid ingredients/nutrients | 3.749 | 1.056 | 1 | 5 |
|  | LBLS | Read labels | 3.823 | 1.001 | 1 | 5 |
|  | LOCA | Find local foods | 3.873 | 1.046 | 1 | 5 |
|  | PRNC | Stand up for principles | 3.764 | 1.018 | 1 | 5 |
|  | CONX | Consuming consciously | 3.987 | 0.939 | 1 | 5 |
| Group 4 | INFO | 1 if looks for information | 0.706 | 0.456 | 0 | 1 |
|  | BIOL | 1 if favours biologic/organic | 0.678 | 0.468 | 0 | 1 |
|  | COOK | 1 if cooks | 0.877 | 0.329 | 0 | 1 |
|  | FAFH | Meals away-from-home | 2.143 | 2.202 | 0 | 10 |
|  | FRTE | Meals ready-to-eat | 1.108 | 1.741 | 0 | 10 |
|  | LFTO | 1 if uses leftovers | 0.550 | 0.498 | 0 | 1 |
|  | SHOP | 1 if the one who buys food | 0.908 | 0.290 | 0 | 1 |
|  | FEXP | Expenditure on food | 5.502 | 2.179 | 1 | 9 |
|  | ROFP | 1 if owns food production | 0.406 | 0.491 | 0 | 1 |
|  | CHNG | 1 if willing to change diet | 0.639 | 0.480 | 0 | 1 |
|  | AWAR | Construct food awareness | 3.688 | 0.639 | 1.07 | 5 |

## 3. Results

Current diets were directly addressed in the questionnaire where participants were questioned: "Which of the following diets do you most identify with?", with a range of answers from: "Omnivorous (consumes every type of foods, animal-based and plant-based)"; "Pescatarian (fish-based, excludes meat)"; "Flexitarian (mainly plant-based, significant reduction in animal-based foods)"; "Ovo-lacto-vegetarian (plant-based, includes animal-based by-products)"; "Vegan (plant-based, excludes all types of animal-based foods)"; "Other: which?". Following the answers from this question and the answers regarding meals per week for each primary food, it was possible to identify each participants' current diets with greater robustness. The robustness check was necessary because the auto-identification showed some inconsistencies when considering the frequency of meals. In fact, some dissonance was observed between the two. Table 4 marks the differences.

**Table 4.** Comparison between auto-diet and true-diet.

| | True-Diet | | | | | |
| --- | --- | --- | --- | --- | --- | --- |
| **Auto-Diet** | **Omnivorous** | **Pescatarian** | **Flexitarian** | **OLVeg.** | **Vegan** | **Total** |
| Omnivorous | 844 | 0 | 0 | 0 | 0 | 844 |
| Pescatarian | 0 | 8 | 20 | 0 | 0 | 28 |
| Flexitarian | 0 | 17 | 72 | 0 | 0 | 89 |
| O-L-Vegetarian | 0 | 9 | 12 | 26 | 0 | 47 |
| Vegan | 0 | 1 | 7 | 6 | 18 | 32 |
| Total | 844 | 35 | 111 | 32 | 18 | 1040 |

From Table 4, when comparing auto-diet and true-diet, one concludes that there is a dissonance between the diet the participant identifies with and the diet they actually practice. For example, of the 47 self-identified as ovo-lacto-vegetarians, in practice, 12 are flexitarian and 9 are pescatarian, going down to only 32. These changes are due to self-identified consumers pointing out sporadic animal-based meals. The same is true for the rest of the non-omnivore categories. Some flexitarians (17 pescatarians) did not report consuming meals with meat. However, since some of the other categories have flexitarians "by mistake", the real number of flexitarians is higher than the self-identified (111 > 89). It is also observed that in practice the number of ovo-lacto-vegetarians and vegans is lower.

This dissonance between "theory" and "practice" is mostly found in participants who sporadically consume foods from another category or who are in a transition phase, i.e., from one category to another. Thus, it is important to make this dissonance effect clear to be considered in future questionnaires. The participant's perception, not due to the inattention likely to exist, but to an intrinsic unawareness, may not translate reality as it is practiced. Hence, it is important to introduce more concrete questions, particularly, the number of meals per type of food, to validate the current diets.

According to a study conducted in Portugal [66], the number of vegetarians (ovo-lacto-vegetarians and vegans) in 2017 represented 1.2% of the national working population. For a vegan diet, the figure was 0.6%. Compared to the present study, the number of both has increased significantly to 3.8% and 1.7%, respectively, as shown in Figure 2. Most respondents (80%) follow an omnivorous diet, whereas the reminiscent is distributed between the alternative diets. Flexitarians reach more than 10%, whereas pescatarians, ovo-lacto-vegetarians, and vegans are below 4% each.

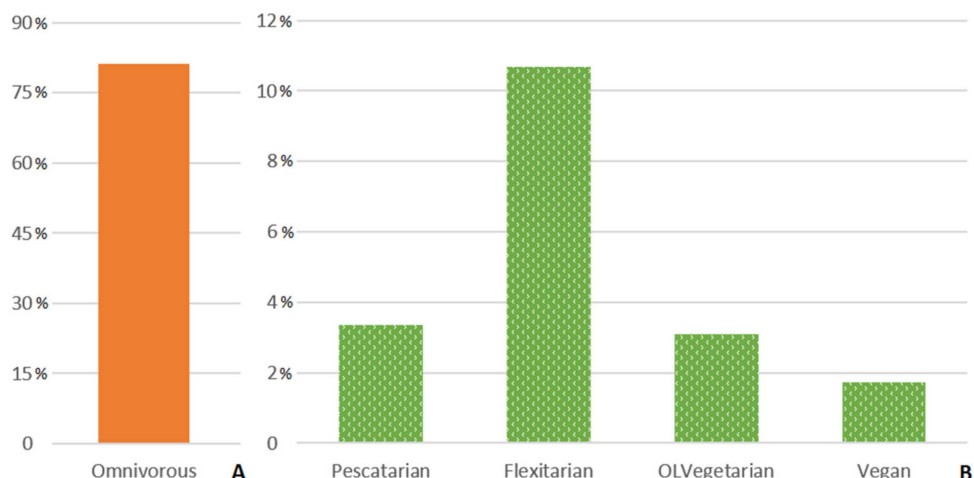

**Figure 2.** Current food choices by diets (%) (panel (**A**) shows the percentage of omnivorous consumers, panel (**B**) shows the percentage of non-omnivorous consumers; both panels have different scales to facilitate reading).

Considering the number of models, to facilitate analysis, Figures 3–6 show the coefficients for each set of models as described in the general specifications (the y-axis represents the variables assessed). Although the value of the coefficients is not to be interpreted since it does not reflect the marginal effect, the overall impact (positive or negative) of the factor can be derived, and, thus, it is possible to understand if the predictor supports or hinders the specific food choice. Additionally, with the marginal effects, in the Supplementary Material (Tables S2–S21), it is possible to quantify the impact of each predictor on each dependent variable, that is, the impact on the probability of choosing zero red meat meals, or the probability of choosing four to six vegan meals per week. The marginal effects regarding the motivators from Group 1 and Group 2 for red meat and vegan meals are analysed through Figures 7–10 (due to space constraints, the figures reporting the marginal effects of all models (Figures S1–S16) are available in the Supplementary Material.). Following the parsimonious principle, the variables which were not significant were excluded from the models. The econometric software STATA 15 was used.

Considering the socioeconomic characteristics (Group 1) depicted in Figure 3, the older the consumer, the more likely he/she will consume fish meals. Body mass index also has a positive impact, but this time is towards red and white meat meals, whereas it is negatively linked with vegan meals. This means that the probability of choosing red and white meat increases as the BMI increases, and vegan meals decrease for the same increase. The income of the household is positively linked to consumers' choice of fish and ovo-lacto-vegetarian meals. Moreover, female consumers are less likely to consume red meat meals (or more likely to consume zero red meat meals, according to the detailed marginal effects) compared to male consumers, and are more likely to consume more vegan meals. On the contrary, a consumer who lives with her/his family, compared to living alone or sharing a house, is less likely to consume vegan meals, whereas it is positively linked with the consumption of red, white, and fish meals. Higher education is associated with fewer red meat meals; however, only at the 10% significance level. Liberal consumers contrast with Catholic ones, where the first are negatively linked with animal-based foods, and the latter are positively linked. Additionally, being Catholic decreases the probability of eating vegan. Being left was not statistically significant for any food choice, whereas right inclination leads to more red meat meals. A student is more likely to eat white meat and vegetarian meals compared to non-students.

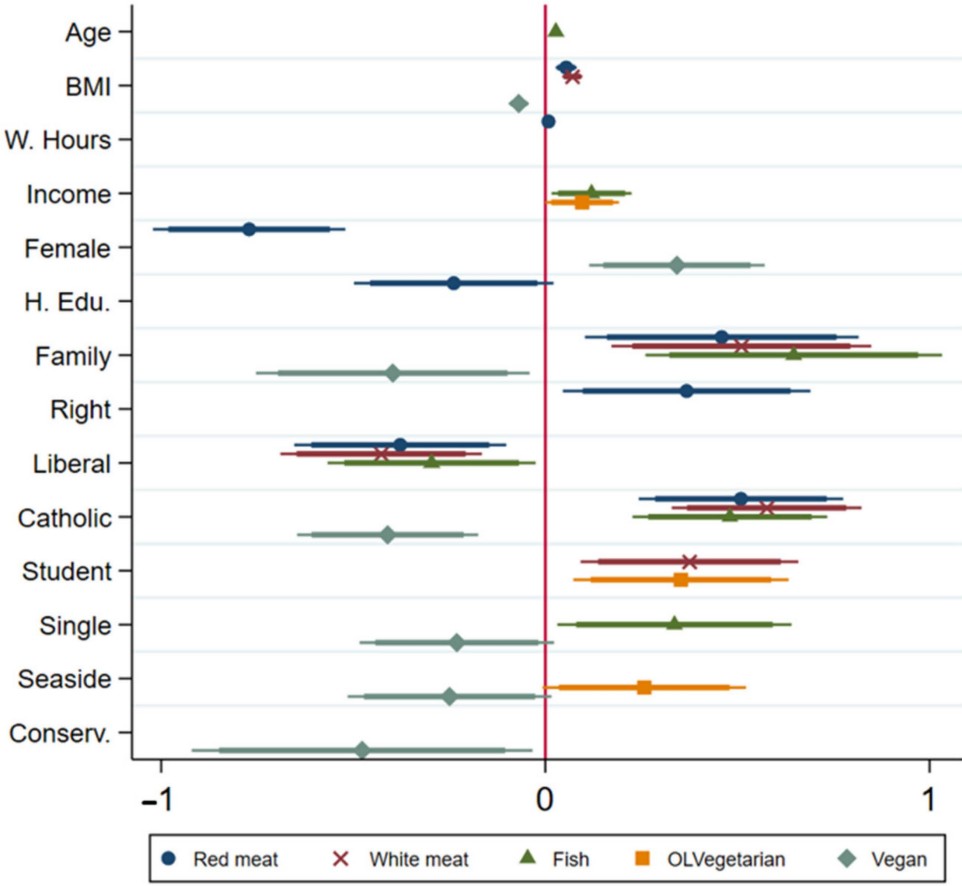

**Figure 3.** Results of socioeconomic characteristics for all food choices. The thick and the thin lines correspond to the 10 and 5% statistical significance levels, respectively. The variable is statistically significant when these lines do not cross the horizontal red line. The x-axis shows the coefficient value.

Following Figure 4, both general orientations (Group 2) appearance and animal wellbeing show to be consistent predictors of animal-based and plant-based meals; however, with opposite effects. Whereas consumers with higher orientations toward appearance are positively linked to animal-based meals, consumers with higher orientations toward animal wellbeing are negatively linked with such meals. For vegan meals, the relationship is inversed. Pleasure and appearance show similar effects meats and vegan meals. Nutrition seems to be a predictor of fish meals (positive) and red meat meals (negative). Higher importance given to the environment is associated with higher quantities of plant-based meals (both vegetarian and vegan), whereas price was only associated with more frequent vegetarian meals, but also white meat meals. For red meat consumption, convenience shows to be a positive predictor. Naturalness, similar to environment, is positively associated with more vegan meals. The importance given to price seems to be only affecting white meat and vegetarian meals, which, in turn, might be cheaper than the other food choices. Valuing status quo is associated with eating more red meat, whereas valuing information shows a negative link (only at 10%). Variety and information are associated with more and less fish consumption, respectively.

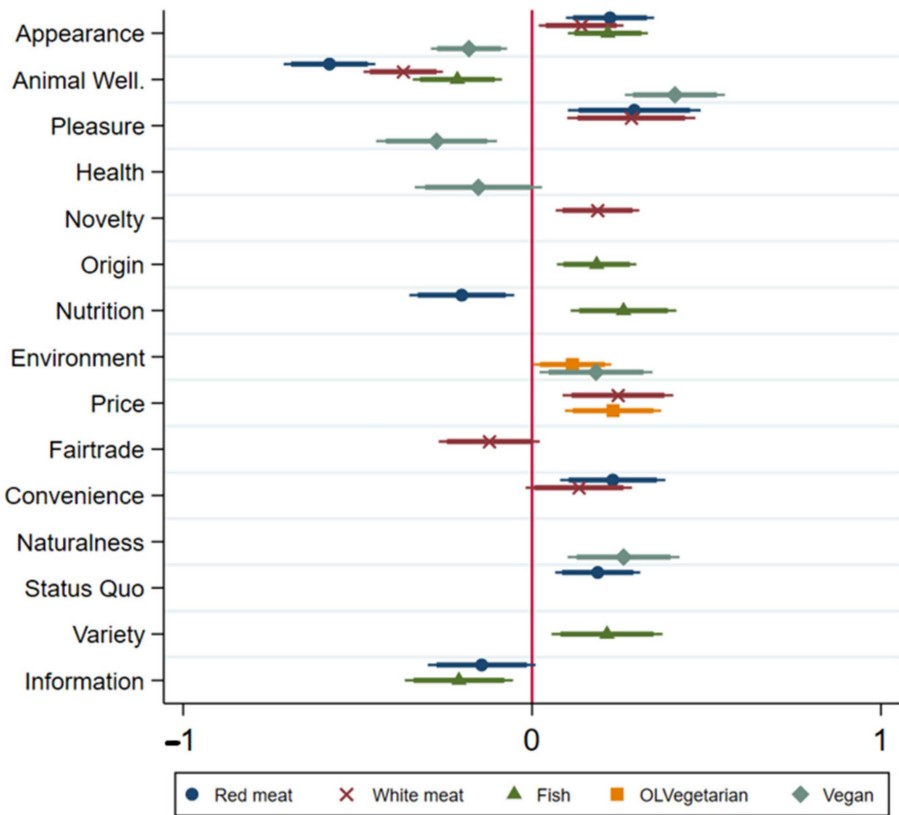

**Figure 4.** Results of general food consumption orientations for all food choices. The thick and the thin lines correspond to the 10 and 5% statistical significance levels, respectively. The variable is statistically significant when these lines do not cross the horizontal red line. The *x*-axis shows the coefficient value.

Regarding specific food consumption orientations and concerns (Group 3) shown in Figure 5, consumers who show deep concern with losing weight are likely to eat fewer vegan meals, while eating more ovo-lacto-vegetarian meals, as well as meat meals. Gaining weight is, in contrast, associated with more vegan meals, although it is also related to white meat. Concern in avoiding certain ingredients and/or nutrients, such as salt or carbohydrates, is associated with fish and white meat meals. Deeper concerns for reading labels and finding local foods are negatively linked with red meat meals and positively linked with vegan meals, respectively. The concerns related to conscious consumption and defending principles through food choices are positive predictors of plant-based meals, and negative for animal-based meals.

As shown in Figure 6, the results suggest that consumers who eat out are more likely to eat more red meat and fish meals, but also vegetarian ones. For ready-to-eat meals, it is negatively linked with fish meals. Higher food expenditures are linked with more red and fish meals, and less vegan ones. For the binary variables, a consumer who cooks for him/her or his/her family is less likely to choose red meat in his/her meals, and more likely to choose plant-based foods instead. The same is observed for favouring biologic/organic foods, looking for information before buying, and willingness to change diets. Consumers who consider these are less likely to eat animal-based foods, and more likely to eat plant-based foods. Additionally, a consumer who buys his/her food and has/receives own production is positively associated with more vegan meals. About the construct concerning consumers' awareness on food-related issues, a consumer which is more aware about these (Table S1) is more likely to choose more plant-based and fish meals, and less likely to choose red meat meals.

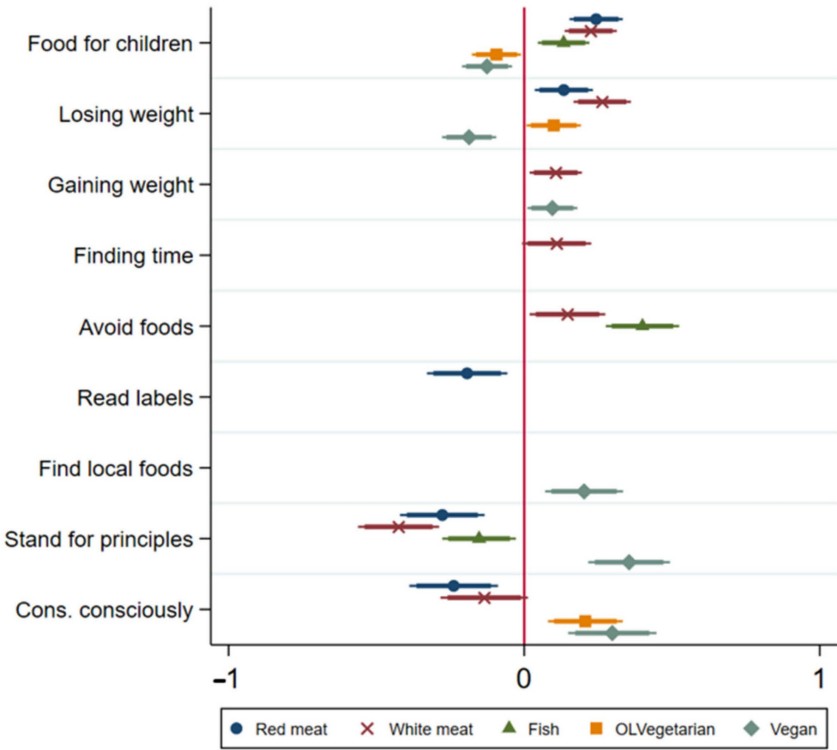

**Figure 5.** Results of specific food consumption concerns for all food choices. The thick and the thin lines correspond to the 10 and 5% statistical significance levels, respectively. The variable is statistically significant when these lines do not cross the horizontal red line. The *x*-axis shows the coefficient value.

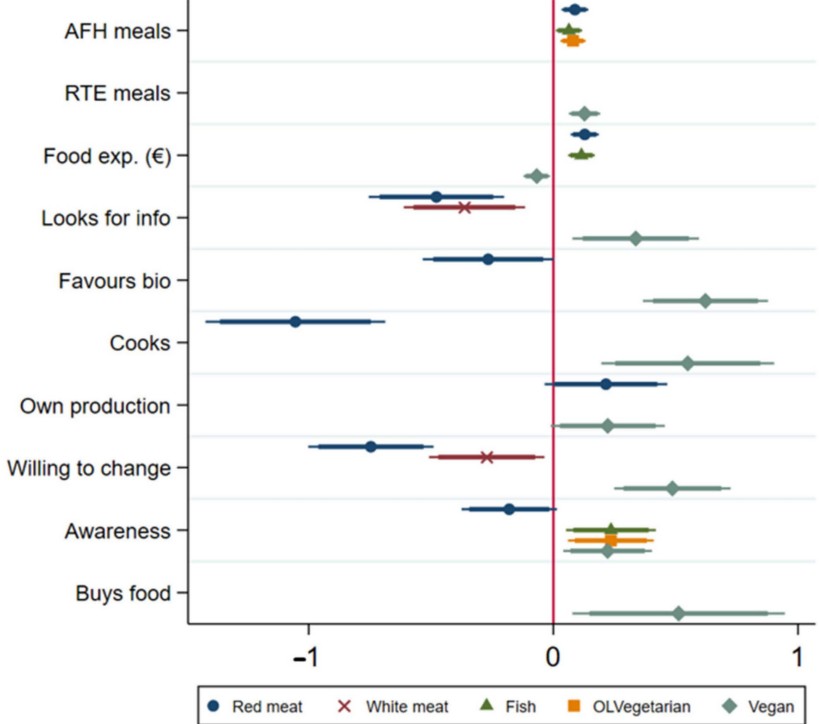

**Figure 6.** Results of food consumption preferences/behaviours for all food choices. The thick and the thin lines correspond to the 10 and 5% statistical significance levels, respectively. The variable is statistically significant when these lines do not cross the horizontal red line. The *x*-axis shows the coefficient value.

Regarding the marginal effects, Figure 7 shows some of the predictors (Group 2) mentioned above for red meat meals. It is possible to interpret that, on average, a one-unit increase in the preference for animal wellbeing (5-point Likert-type scale) is associated with a 0.084 increase in the probability of having zero red meat meals per week (a) and a 0.02 increase in having one to three meals (b). This is offset by a decrease of 0.087 in the probability of having four to six meals (c) and a 0.017 decrease in having seven or more red meat meals per week (d) (Table S7). The same can be said about pleasure, but with the opposite effect. Following a one-unit increase in importance, the probability of eating 0 and 1–3 meals decreases by 0.021 and 0.018, respectively, and of eating 4–6 and 7–7+ meals increases by 0.032 and 0.008 (Table S7).

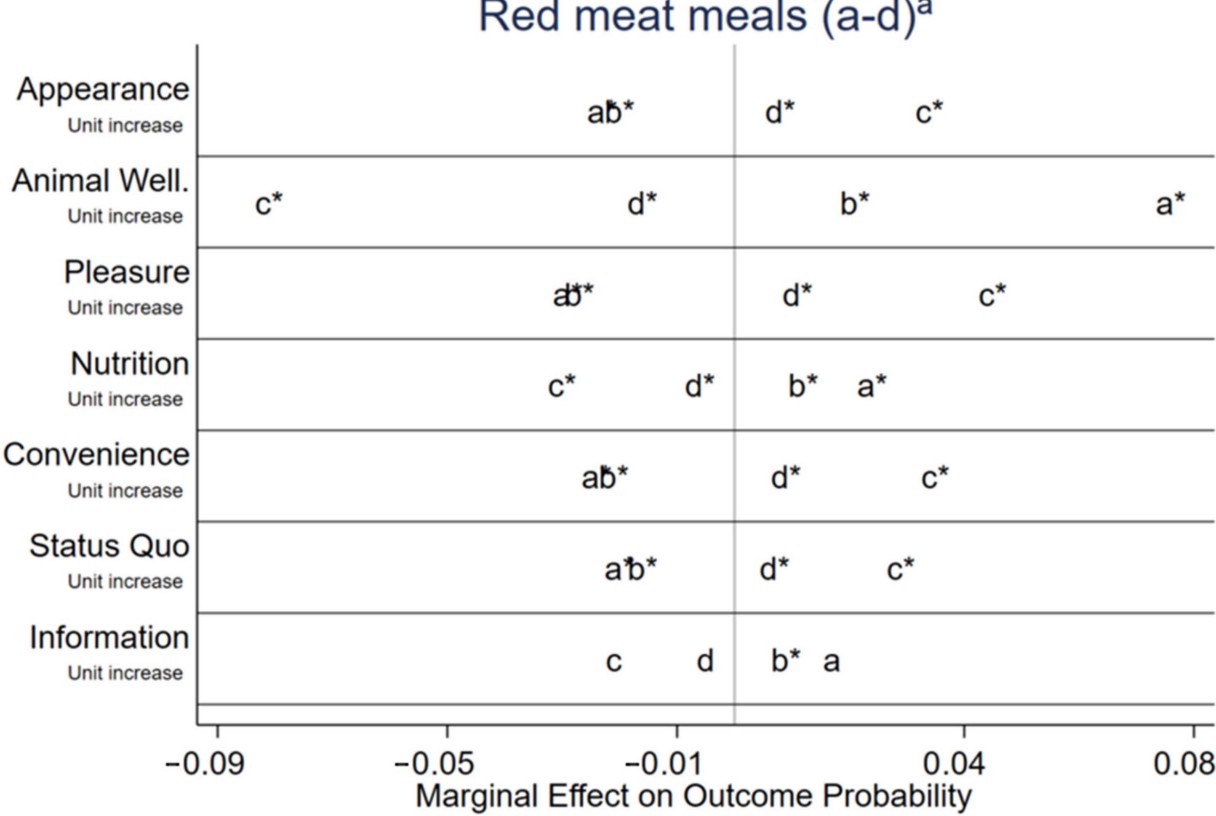

**Figure 7.** Marginal effects of general food consumption orientations on red meat meals. The * denotes statistical significance at the 5% level.

Moreover, Figure 8 depicts the same group of variables, but for vegan meals. The contrast in the marginal effects between vegan meals and red meat meals is clear, particularly, for appearance, animal wellbeing, and pleasure. On average, following a one-unit increment in the importance given to pleasure, the probability of eating seven or more vegan meals decreases by 0.05 and 0.022 for four to six meals (Table S11). The probabilities of eating lower quantities increase. Regarding animal wellbeing, it is where the highest impacts are shown. A one-unit increase is associated with a 0.061 increase in the probability of choosing seven or more plant-based meals, and a 0.065 decrease for choosing zero meals (Table S11).

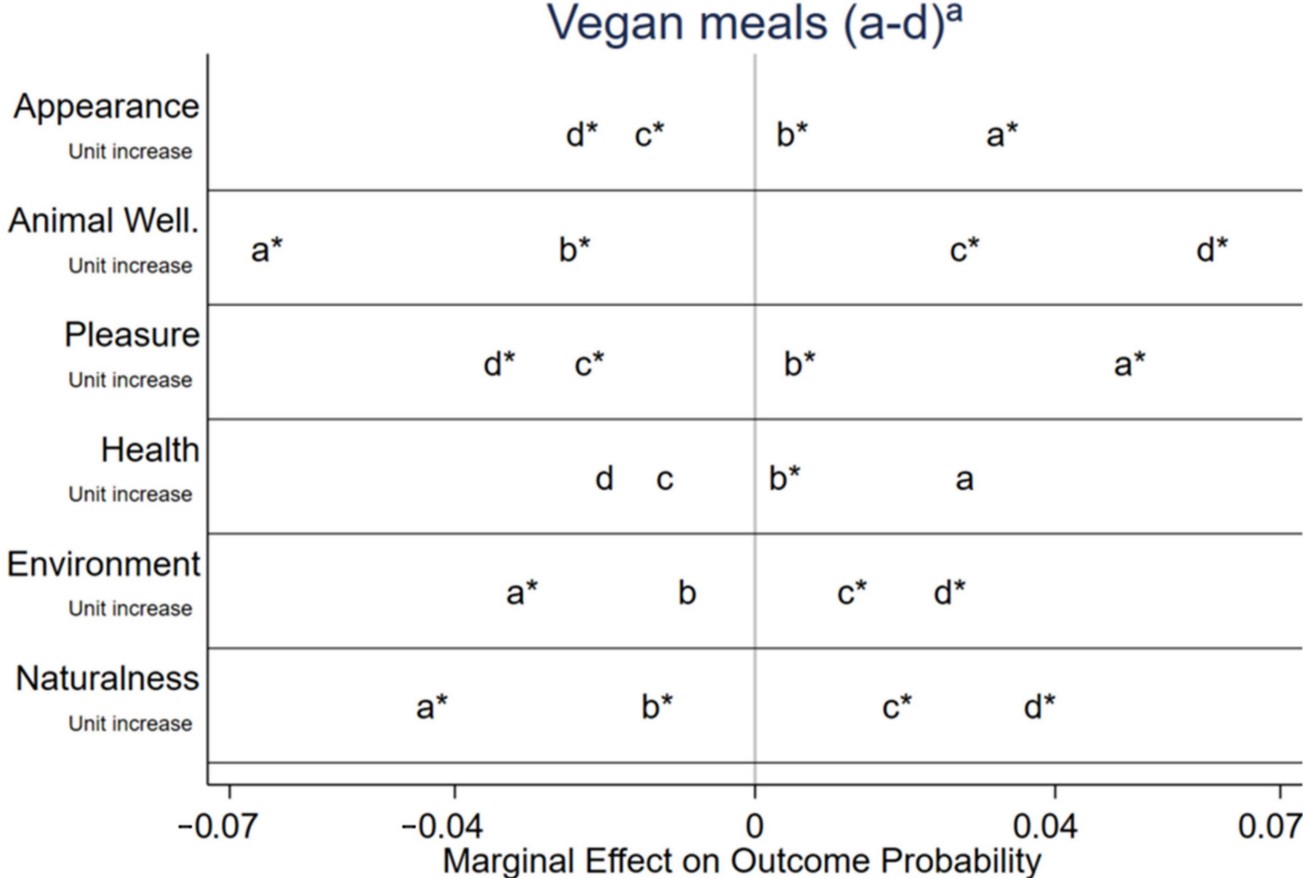

**Figure 8.** Marginal effects of general food consumption orientations on vegan meals. The * denotes statistical significance at the 5% level.

Considering the socioeconomic variables for red meat meals from Figure 9, the predicted probability of consuming zero meals is, on average, 0.085 higher for female consumers than for an otherwise similar male consumer, whereas the predicted probability of consuming four to six meals is 0.125 lower (Table S2). On the contrary, living with family increases a consumer's probability of eating four to six meals and seven or more by 0.069 and 0.014, respectively, compared with a consumer living alone or sharing a house with someone other than family. The predicted probability of consuming one to three red meat meals is 0.026 lower, and 0.058 lower for zero meals (Table S2). Considering the continuous variable BMI, a standard deviation increase in BMI (about 4.53, Table 3) is associated with a 0.026 decrease in the probability of eating zero meals, whereas the probability increases 0.04 for four to six meals, including red meat.

The same interpretations can be derived for vegan meals, following Figure 10. The predicted probability of consuming the highest frequency of vegan meals is, on average, 0.047 higher for female consumers than for an otherwise similar male consumer, whereas the predicted probability of consuming zero meals is 0.063 lower (Table S6). In contrast, living with family decreases a consumer's probability of eating seven or more vegan meals by 0.06. The predicted probability of consuming zero meals is 0.068 higher (Table S6). Considering the continuous variable BMI, a standard deviation increase in BMI is associated with a 0.061 increase in the probability of eating zero meals, whereas the probability decreases by 0.039 for seven or more plant-based meals.

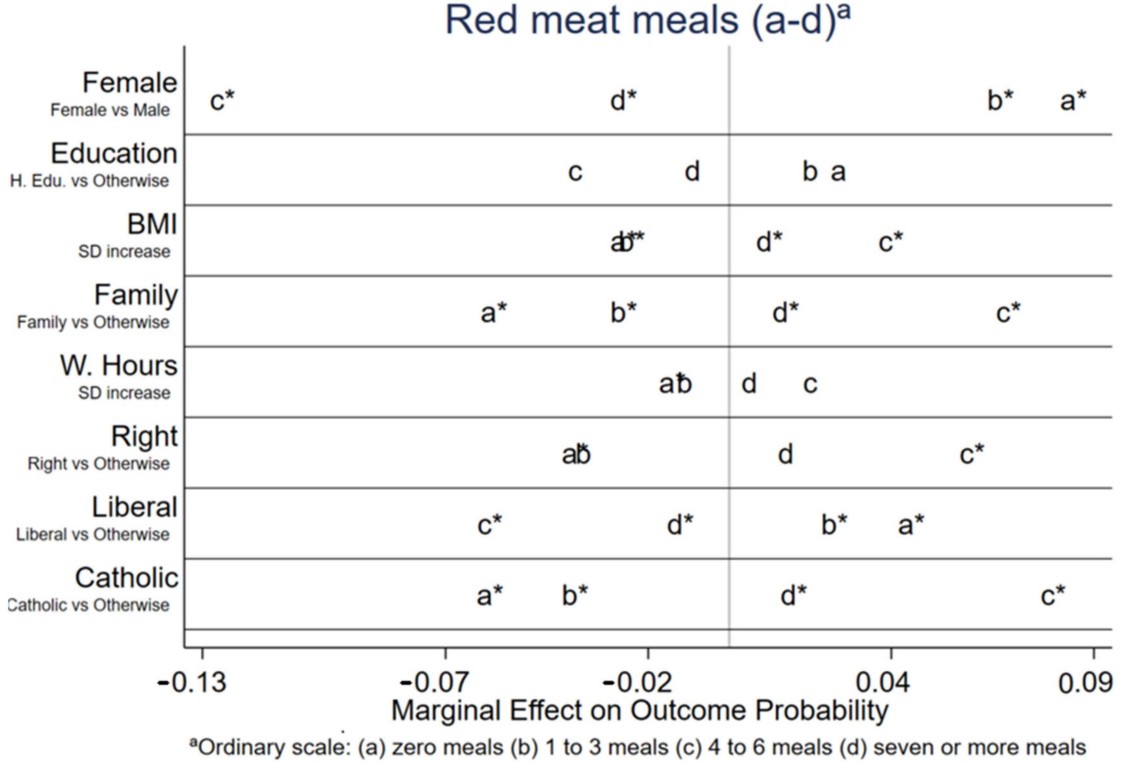

**Figure 9.** Marginal effects of socioeconomic characteristics on red meat meals. The * denotes statistical significance at the 5% level.

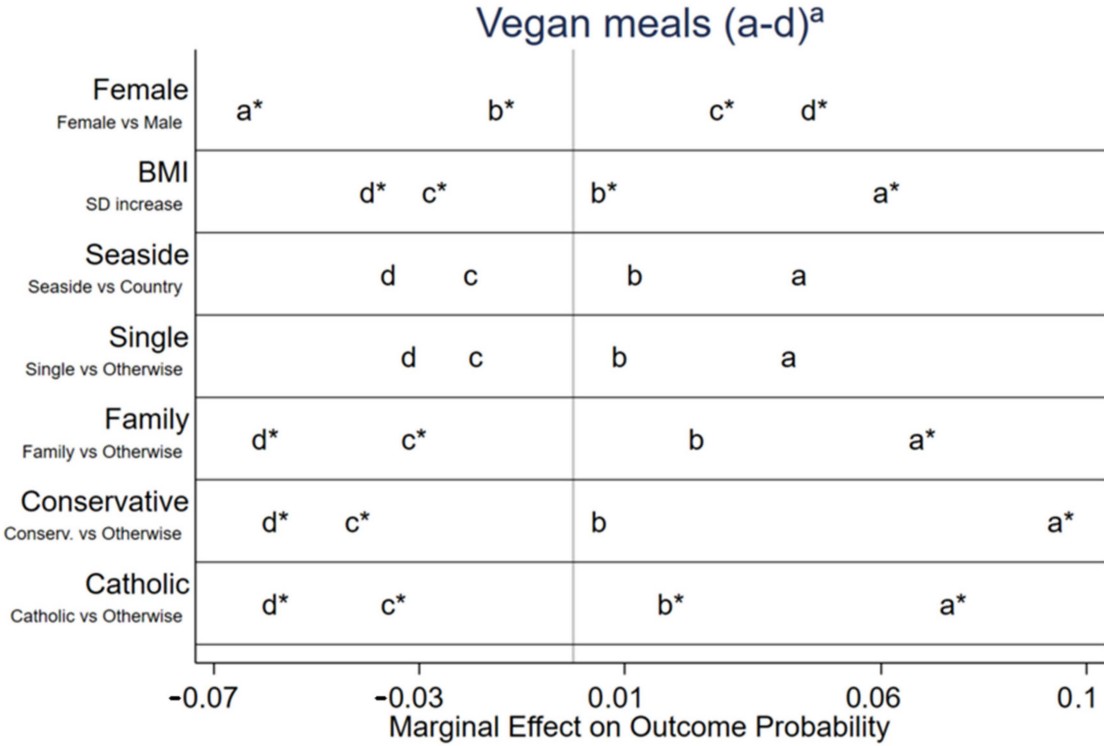

**Figure 10.** Marginal effect of socioeconomic characteristics on vegan meals. The * denotes statistical significance at the 5% level.

In short, the main findings suggest that plant-based meals are primarily motivated by factors regarding animal welfare, the environment, and naturalness, corroborating Neff et al. [67] and Graça et al. [62], and that conscious, more informed, and aware consumers tend to eat more plant-based meals. These motivations and consumer traits were also associated with fewer animal-based meals. Additionally, consumers who are responsible for shopping and cooking their meals also tend to choose more plant-based meals. Favouring biologic/organic, local products, and willing to change diets are also associated with more plant-based meals, and also related with fewer animal-based meals, in agreement with Cliceri et al. [68]. Female, more educated, and higher income consumers tend to eat less meat and more plant-based meals, strengthening the insights from Pfeiler and Egloff [69] and Rozin et al. [70]. In general, most of the drivers of plant-based meals are also motivators behind the reduction of animal-based meals. Among the socioeconomic and demographic aspects, it seems that living with family promotes a higher frequency of animal-based meals and lower for plant-based meals.

## 4. Discussion

Considering the importance of understanding how to promote healthy and sustainable food choices, particularly, a reduction in animal-based meals followed by an increase in plant-based meals, which goes according to the literature, the present study assesses the impact of different potential drivers behind food choices. According to the results, several motivators have the capacity to reroute current food choices if targeted effectively. The policy implications described can be generalized for other European countries, particularly, southern ones with similar culture and where the Mediterranean diet is highly promoted.

*Food Policy Implications on Promoting Healthy and More Sustainable Food Choices*

Due to market power, externalities, and imperfect information, markets are prone to market failures, and, particularly, the food market [71]. Following the welfare state theory, policymakers should intervene in the markets to mitigate any chance of potential market failures. Since the food sector endorses a variety of externalities, it is a plausible target for regulation. Considering the insights revealed, it is recommended that food policy should be focused on promoting higher levels of information regarding food, their characteristics, and multiple effects. This way consumers have the necessary information, particularly, the environmental and health externalities, to make informed food choices. An example of guaranteeing full information on the externalities is that these, both positive and negative (through subsidies and taxes, respectively), be incorporated in the final price. Reading labels, looking for information before buying, general orientations towards information, and overall awareness high scores about food issues negatively affect the consumption of red meat meals, while contributing to the promotion of plant-based meals, as summarized in Figure 11. This insight reveals that consumers who are "involved" [72] and search for information, and ultimately are more informed, tend to eat healthier and more sustainable. The same can be said about conscious consumers, as the results show. General orientations towards the environment, preferences for local and biologic/organic foods, concerns for defending his/her principles, and consuming consciously negatively affect the consumption of red meat, while promoting plant-based meals.

Nevertheless, supplying more information alone might not be as effective as it might appear. According to a report from Allied Market Research, the global market for animal-protein substitutes will reach $7.5 bn by 2025 [73]. In the past few years, the current supply of plant-based foods and information about them has skyrocketed [74], with novel food products flooding the market, but also with more campaigns and information materials being developed and available to the public.

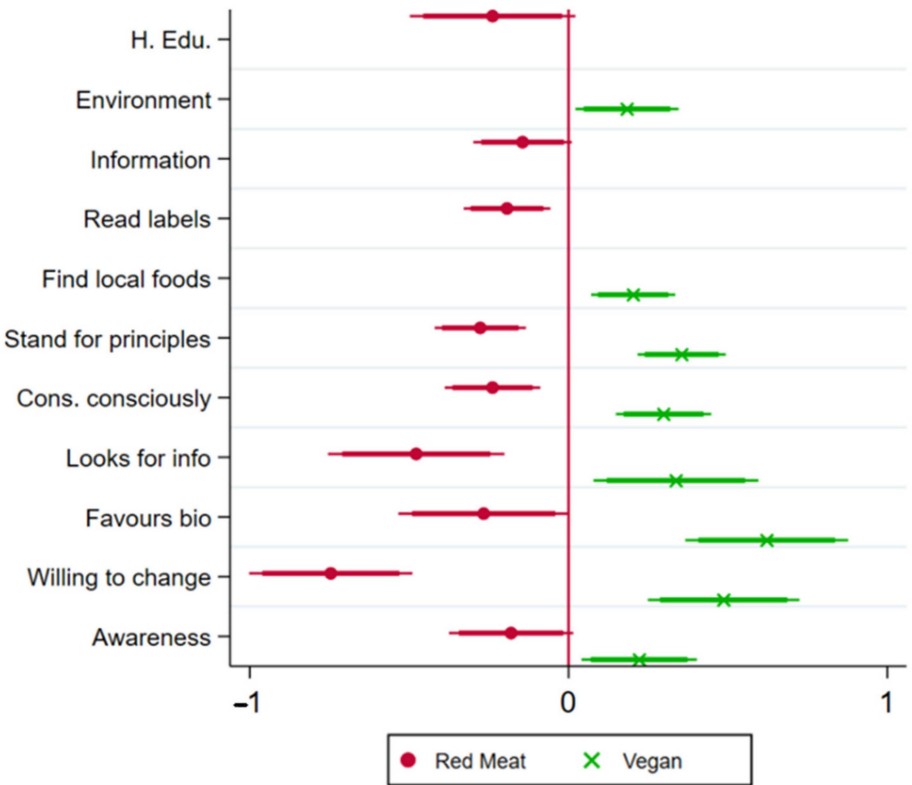

**Figure 11.** Effects of conscious and information motivators on food choices. The thick and the thin lines correspond to the 10 and 5% statistical significance levels, respectively. The variable is statistically significant when these lines do not cross the horizontal red line. The *x*-axis shows the coefficient value.

Following Galli et al. [17], in Portugal, the information gap which the authors identify is of no major concern, as strategies to achieve better information and awareness campaigns are present. Thus, information per se might not be the most effective solution, since consumers might not assimilate it and use it effectively. The current findings suggest that not only do informed consumers follow healthier and more sustainable diets, but more importantly, so do consumers who actively look for information before choosing what to consume, i.e., consumers who have rooted the habit of staying up to date and follow behaviours that maintain that level of knowledge regarding food. Edenbrandt and Lagerkvist [54] conclude that carbon labels affect consumer choices towards lower-emitting foods, but that the largest effect is among consumers who currently purchase sustainably. Additionally, Fang et al. [75] found similar evidence, i.e., nutrition labels mainly affect consumers who already use health labels. This might suggest that more information indeed could be an effective food policy to promote healthier and more sustainable food choices, but only for consumers who already have the habit of using information materials and are willing to change current choices. A portion of consumers who are not yet looking for and making healthier and more sustainable choices, and, consequently, the ones who might have the highest mitigation potential, might be left out of reach of these policies.

Edenbrandt and Lagerkvist [54] emphasize that their analysis is "based on the premise that the consumers are informed about the meaning of the label, and their attention is focused toward the label". Although more information can lead to effects other than the ones expected from consumers, such as a reformulation of products from the production-side as it happened in Portugal [76], if consumers are not in the habit of seeking information, policies surrounding this measure can be effective, although inefficient. Therefore, the focus of policymakers, instead of guaranteeing clear and accessible information to all, which is necessary, should be redirected to education and create the habit of routinely seeking information and acquiring knowledge regarding food.

Market-based instruments such as taxes and subsidies, and information-based policies such as campaigns and labelling can have potential impacts in changing behaviour [42]. However, the changes required are structural, and need to deal with the current rooted dietary habits characterized by overconsumption and the disregard for the externalities associated with current food choices. A consumer living with her/his family is more likely to choose more animal-based meals and less likely to choose plant-based ones (Figures 9 and 10). This could be showing evidence of the influence of the deep cultural roots regarding family tradition and habits. This effect could be somehow rerouted following a strong food policy initiative in schools. Education regarding food and its vast branches should be mandatory in all levels of the educational systems, starting at the kindergartens to high school and further higher education institutions.

As spaces for the formation of individuals, education institutions should increasingly assume the need to provide their students with personal and social skills that allow them to have an active voice in society. This investment in the formation will only be fully realized with healthy and conscious citizens throughout their life, living in a healthy and habitable planet with opportunities and not barriers. Thus, the importance of health and sustainability issues as a central part of these institutions' mission, not only for students, but also for the entire community, particularly, on a key component of an individual's life such as food choices. The value-action gap—the tendency to express pro-environmental orientations without coupling them with pro-environmental behaviour—is a persistent barrier for policymakers. Encouraging sustainable dietary habits from an early age can help bridge this gap [77]. Thus, the youth have the opportunity to change current and future behaviours by disseminating the information learned at school, influencing their peers and the household in the present, and affecting the future, since today's youth are the future's consumers and policymakers.

Concerning the importance of education, Pais et al. [78] highlight the example of school gardens and the spillover effects that come from it: from the positive "contamination" of good habits into their households, to narrowing the gap between humans and nature, and also the exploitation of school gardens in teaching subjects such as mathematics. In urbanistic terms, food gardening has received increased interest as a tool to promote sustainable urban development [79]. Kovacs et al. [80] recommend holistic food policies such as the provision of healthy (and sustainable) foods, lowering the supply of unhealthy (and unsustainable) foods, restricting marketing, and promoting education. The authors further highlight the need for consistent indicators of progress to evaluate the effectiveness. These indicators could be built and monitored through questionnaires such as the present one.

In Portugal, policymakers have approved the restriction of advertising regarding foods and beverages with higher sugar levels, fats, and salt within a 500-m radius from schools, and in TV programs directed to children (Decree Law No. 330/90, 2019). However, strong political commitment to shift diets is still lacking, as Galli et al. [17] conclude. Despite education as a vital tool to change dietary habits of future consumers, the present ones, which currently make food choices also need to be addressed. To promote healthier and more sustainable food choices, both local actions, as well as national strategies, are required. The authors suggest the inclusion of sustainability considerations in the development of dietary guidelines to re-orient current food choices [17,81]. Van Loo et al. [72] reinforces that dietary guidelines combining both health and sustainability aspects is vital and should be well received by consumers, so healthy, sustainable, and plant-based diets are closely matched and not perceived as conflicting. Additionally, Temme et al. [40] discuss further government demand-side food policies from market-based, information-based, and behavioural policies. Teaching conscious consumption in the era of consumerism and sowing the view that individual food choices can have an impact on global trends, within a reformed and tailored educational system, can have potentially positive effects in not only changing behaviours, but building them, both for current and future consumers.

Nevertheless, it is important to note that conscious and informed consumers do not necessarily mean consumers with higher levels of education. Although the findings suggest that consumers with higher education are more likely to eat fewer red meat meals, it is only at the 10% statistical significance level. Moreover, higher education seems only to affect red meat, despite the literature suggesting it as an enabler for following more plant-based diets [46,69,82]. Thus, it can be concluded that it might not be as relevant to have a higher degree than for the individual to seek for information and consume consciously, a trait that can be built within mandatory school years and through local actions, such as information campaigns, and national strategies, such as dietary guidelines. Kirbiš et al. [83] further reinforce the importance of public health and environmental campaigns focused on the less-educated groups. The authors show that these groups are significantly less likely to hold sustainable attitudes. Market-based instruments are also effective mechanisms to reroute current food choices towards healthier and more sustainable ones.

Regarding another potential driver of food choices, price does not negatively affect any of the food choices, despite the literature suggesting higher meat prices as a trigger to reduce meat consumption [84]. In contrast, plant-based foods, particularly, plant-protein substitutes, might be generally seen by the public as more expensive than their animal counterparts, and, consequently, plant-based diets as more expensive. However, the literature suggests food budget savings from being vegan, ranging from 7 to 15% [85–87]. A recent study assessing Portuguese consumers demonstrated that consumers who follow a plant-based diet spend less than their omnivorous counterparts [88]. According to the results, price is associated with white meat and vegetarian meals, which might be cheaper than red meat and fish meals. Nonetheless, consumers who spend more on food tend to consume more red meat and fish meals, whereas the effect is negative for vegan meals. Additionally, the food retailers and the industry can also benefit from the findings presented, as they can better define tailored marketing strategies for targeting specific segments of consumers. However, food retailers might not be interested in promoting healthier and more sustainable diets if not profitable, which might conflict with the work of policymakers. Nonetheless, retailers continue to expand their selection of natural, organic, non-genetically modified, and other alternative foods, which could be highly beneficial to achieve the transition proposed.

Finally, concerning the socioeconomic determinants (Figure 3), the seminal work of Gossard et al. [89] demonstrate that gender affects dietary habits, which corroborates with the current results. The latter authors show that women consume substantially less than meat than men. The gender differences in dietary habits could be explained in part by differences in the values of men and women, as eating meat could be associated with "manhood" [90,91]. More recent survey-based studies have shown similar conclusions, particularly, Clonan et al. [92] and Koch et al. [82], which suggest that women do consume less animal-based foods than men.

## 5. Conclusions

Rerouting current food choices is recognized as a fundamental challenge to guarantee the sustainability of the planet and its population. To materialize these changes, policymakers and other market agents need to develop strategies that are effective and efficient. Understanding the motivations (drivers and barriers) behind consumers' food choices is essential to develop such strategies. Thus, this study aimed to provide inputs to better inform and design these strategies and policy guidelines.

Following a nationally representative sample (N = 1040), a total of 51 motivators were assessed regarding their effects on food choices. Globally, most of the drivers of plant-based meals were also enablers for the reduction of animal-based meals. One of the main findings suggest that not only is promoting clear and accessible information important, but guaranteeing that consumers actively seek out information and know how to use it should be the focus of food policies. Education about food should be mandatory to disseminate knowledge more effectively and to build, and not only change to, healthier and more sus-

tainable food choices. The lessons from Portugal described can be useful for countries with similar dietary habits such as the Mediterranean diet commonly followed in South European countries.

**Supplementary Materials:** The following supporting information can be downloaded at: https://www.mdpi.com/article/10.3390/su15043868/s1, How to promote healthier and more sustainable food choices: The case of Portugal.

**Author Contributions:** Conceptualization, D.F.P.; Methodology, D.F.P.; Validation, A.C.M. and J.A.F.; Investigation, D.F.P.; Data curation, D.F.P.; Writing—original draft, D.F.P.; Writing—review & editing, A.C.M.; Supervision, A.C.M. and J.A.F. All authors have read and agreed to the published version of the manuscript.

**Funding:** The author, Daniel Francisco Pais, is grateful for the PhD scholarship SFRH/BD/143658/2019 funded by the *Fundação para a Ciência e a Tecnologia, I.P.* (FCT, IP). The financial support of the NECE-UBI, Research Unit in Business Science and Economics, sponsored by the Portuguese Foundation for the Development of Science and Technology, project UID/GES/04630/2020, is acknowledged.

**Institutional Review Board Statement:** The study was conducted in accordance with the Declaration of Helsinki, and approved by the Ethics Committee of University of Beira Interior (protocol code CE-UBI-Pj-2021-006 and date of approval 9 February 2021).

**Informed Consent Statement:** Informed consent was obtained from all subjects involved in the study.

**Data Availability Statement:** Additional data is available in the Supplementary Materials. The data base is unavailable due to privacy restrictions.

**Acknowledgments:** The contribution of the three anonymous reviewers and the academic editor is also much acknowledged.

**Conflicts of Interest:** The authors declare no conflict of interest.

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
