# Peer review of "How to Promote Healthier and More Sustainable Food Choices: The Case of Portugal"

_sustainability, doi:10.3390/su15043868_

Round 1
Reviewer 1 Report
The paper is not clearly written and it is difficult to comprehend how the study was performed, how the results are interpreted, what are the main important findings, and what the conclusion is. The Discussion is not a discussion of own results but rather a litterature review, and the Conclusion is weak in that it is only a general consideration very weakly linked to the present study. The impact and novelty of the study appears to be limited.
General comments:
L38-44: Please support these statements with numbers and include a graph illustrating the rapid increase in animal-based consumption.
L45-48: How does eating meat contribute to global warming?
L59-60: Please clarify this statement.
L61: Spell out IPCC and add reference to the report
L72-75: So, in conclusion, all meat is carcinogenic to humans. Elaborate on this statement, especially how it is affected by level of intake and how big is the risk.
L157-159: Delete
Footnote 2: Add translated copy as supplementary material.
Footnote 4: This material is not available for review.
L200- 201: Spell out BMI, HRS, etc.
Table 2 need more explanation and clarification.
Table S1 is not available for review
Table 3: Explain the variable "status quo"
L216: Explain and spell out VIF
L251: Text says 2.4%, but Fig. 1 shows <2%??
Footnote 5: Move to figure legends
Footnote 6 and 7: Add as supplementary material
Reviewer 2 Report
Dear Authors,
This is an example of a very well-planned work with clear information. Results and discussion are appropriate, and the conclusion clearly shows the objectives of the research. However, I recommend changes to improve the quality of work.
Please consider the following:
Abstract
1. The objective is not related to the title. I consider that both should be checked.
2. The abstract should be informative and include the main findings.
3. The motivation of the work should be improved.
Motivation
4. Improve the aim to match the title of the article
Data & Methods
5. A statistical analysis section can be incorporated: describe the design, software…
Results
6. Please check the y-axis of all figures (Especially figure 1)
Conclusion
7. The author must concise the conclusion part as it appears to be too large
Reviewer 3 Report
The Manuscript is well written and presents an important direction towards the reduction of pollution and overall better food sustainability. The methods and sample size were appropriate. the results are presented in easy-to understand manner. And discussion led unambiguously towards conclusions based on obtained results.
lines 59-60: please clarify
lines 90-92: please clarify
Round 2
Reviewer 1 Report
This paper is still a mess and not comprehensive. This paper lacks of novelty, very little impact. Additionally, the authors rebuttal does not adress the comments. Let me give an example. The authors claim in their paper that EATING meat contributes to global warming. This statement prompted a comment. The authors then produces a lenghty answer on how the PRODUCTION of meat contributes to global warming, which we can agree is a different story. However, they did not change their original statement in the revised manuscript. Some other statements are not supported by scientific evidence, and the results are only slightly discussed in the context of previous scientific litterature.
